# Bistability in fatty-acid oxidation resulting from substrate inhibition

**Fentaw Abegaz**[1,2], **Anne-Claire M. F. Martines**[1], **Marcel A. Vieira-Lara**[1], **Melany Rios-Morales**[1], **Dirk-Jan Reijngoud**[1], **Ernst C. Wit**[2,3], **Barbara M. Bakker**[1] *

**1** Laboratory of Pediatrics, Section Systems Medicine of Metabolism and Signaling, University of Groningen, University Medical Center Groningen, Groningen, The Netherlands, **2** Statistics and Probability Unit, University of Groningen, Groningen, The Netherlands, **3** Institute of Computational Science, Università della Svizzera italiana, Lugano, Switzerland

* b.m.bakker01@umcg.nl

## Abstract

In this study we demonstrated through analytic considerations and numerical studies that the mitochondrial fatty-acid β-oxidation can exhibit bistable-hysteresis behavior. In an experimentally validated computational model we identified a specific region in the parameter space in which two distinct stable and one unstable steady state could be attained with different fluxes. The two stable states were referred to as low-flux (disease) and high-flux (healthy) state. By a modular kinetic approach we traced the origin and causes of the bistability back to the distributive kinetics and the conservation of CoA, in particular in the last rounds of the β-oxidation. We then extended the model to investigate various interventions that may confer health benefits by activating the pathway, including (i) activation of the last enzyme MCKAT via its endogenous regulator p46-SHC protein, (ii) addition of a thioesterase (an acyl-CoA hydrolysing enzyme) as a safety valve, and (iii) concomitant activation of a number of upstream and downstream enzymes by short-chain fatty-acids (SCFA), metabolites that are produced from nutritional fibers in the gut. A high concentration of SCFAs, thioesterase activity, and inhibition of the p46Shc protein led to a disappearance of the bistability, leaving only the high-flux state. A better understanding of the switch behavior of the mitochondrial fatty-acid oxidation process between a low- and a high-flux state may lead to dietary and pharmacological intervention in the treatment or prevention of obesity and or non-alcoholic fatty-liver disease.

## Author summary

Obesity is a complex disease which is still poorly understood at the systemic level. Impaired capacity in fat oxidation may contribute to the development of obesity. In this manuscript using a detailed mitochondrial fatty acid oxidation computational model, we demonstrate that the oxidation of fat can exhibit bistability and hysteresis, implying that there is a risk that the pathway gets trapped in a stable state with low activity. We also identify the cause of bistability in the metabolic network and analyse which clinically relevant factors shift the pathway between the high-flux, healthy and low-flux, diseased state.

**Data Availability Statement:** All relevant data are within the manuscript and its Supporting Information files.

**Funding:** This study was funded by the Netherlands Organization for Scientific Research and DSM Nutritional products in the framework of the Complexity programme (grant 645.001.001/3501) (BMB, ECW, and FA) and the Carbokinetic programme (grant ALWCC.2015.6b) (BMB and MRM). ACMFM and MVL received a PhD fellowship from the University Medical Center Groningen. The funders had no role in study design, data collection and analysis, decision to publish, or preparation of the manuscript.

**Competing interests:** The authors have declared that no competing interests exist.

## Introduction

Bi- or multistability, the property of a system to have two or more stable steady states, is widespread in biology. Bistability allows phenotypic variability among cells, despite identical environmental conditions and genotype. This phenomenon has been studied extensively in the context of cell fate switches across all biological kingdoms [1]. Often a positive or double-negative feedback loop in a gene-regulatory or signalling network lies at the origin of the bistability [2–6]. This is for instance the case in the sporulation switch of *Bacillus subtilis* [7], the lactose utilisation network of *Escherichia coli* [8], the eukaryotic cell cycle [9,10] or *Xenopus* oocyte maturation [11].

It is less recognised that also negative feedforward regulation or substrate inhibition may cause bistability [12,13]. In an elegant study, Markevich *et al.* [14] describe how distributive enzyme kinetics in combination with moiety conservation leads to bistability in a mitogen-activated protein kinase (MAPK) signalling pathway. In a distributive mechanism an enzyme catalyses a number of subsequent reaction steps, yet releases the intermediate product(s). For the intermediate to bind again to the active site, it must compete with its precursor. Thus, the precursor acts as a competitive inhibitor and a substrate-inhibition or negative feedforward loop is formed.

Also in metabolic pathways positive feedback loops leading to bistability have been found, particularly in glycolysis [15–17]. Positive feedback loops in glucose metabolism were identified as a cause of population heterogeneity in microbial cultures [18,19]. Substrate inhibition cycles, in contrast, are more likely to occur in lipid metabolism, since repetitive cycles of lipid metabolism are catalysed by the same enzyme(s). Fatty-acid synthase is a processive enzyme: it retains intermediate products, thus precluding inhibition by substrates upstream in the pathway. Fatty-acid oxidation enzymes are, however, at least in part distributive, potentially leading to substrate inhibition [20]. Moreover, since all the pathway intermediates are covalently linked to coenzyme A (CoA), their concentrations are constrained by the conservation of total CoA. Together, these properties may give rise to bistability, just as in the above-mentioned MAPK pathway [14].

In mammalian cells, fatty acids are predominantly degraded via the mitochondrial fatty acid oxidation (mFAO) pathway, also referred to as β-oxidation [21]. This pathway produces ATP from stored fat during times of increased energy demand and low glucose supply, such as fasting, cold exposure or exercise [22–25]. Defects in the mFAO pathway are associated to various metabolic diseases such as obesity or inheritable fatty-acid oxidation disorders [23,24,26–28]. To tackle obesity, enhancing cellular energy expenditure via increasing mFAO and its downstream pathways is considered to be a potential therapeutic strategy [28–33].

A detailed scheme of the mFAO pathway [21,34] is shown in Fig 1. In the cell, long-chain fatty acids are first activated by the formation of acyl-CoA and then transported into the mitochondria by the carnitine shuttle. In each round of oxidation an acyl-CoA molecule is shortened by two carbon atoms, and the latter are bound to free CoA to form acetyl-CoA. An mFAO round consists of four enzymatic reactions and generates in addition one molecule of NADH and one molecule of $FADH_2$. As illustrated in Fig 1, each enzyme catalyses the conversion of several substrates of different chain length. Thus, longer (upstream) and shorter (downstream) metabolites compete for the same enzyme and generate feedforward and feedback inhibition in the pathway [20].

Various kinetic models have been constructed in order to get insight into the dynamics of the mFAO process [20,35,36]. The computational model developed in [20] for rat liver mFAO and further analysed in [37,38] explicitly includes the competition between different substrates

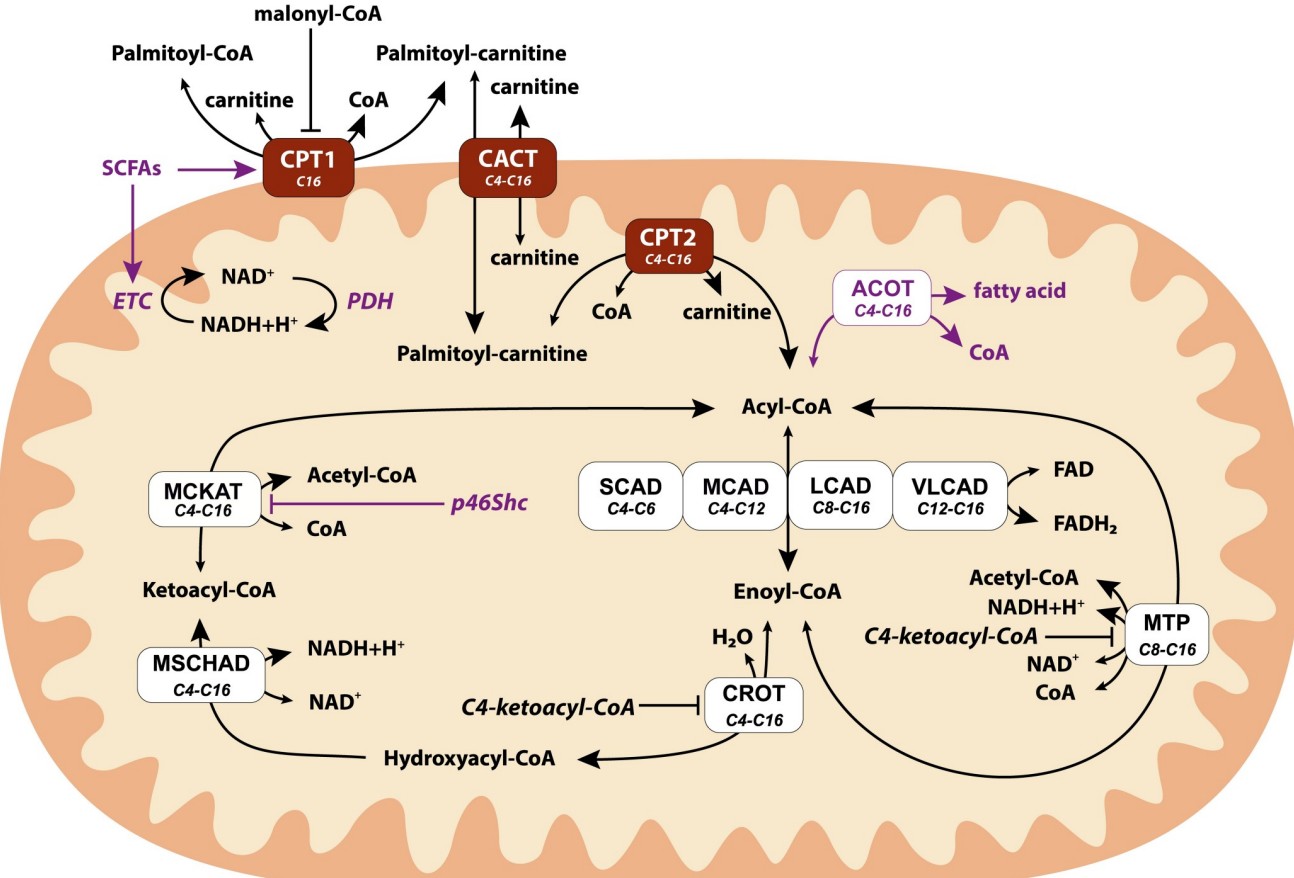

**Fig 1. The mitochondrial fatty acid oxidation pathway as it modelled.** Different colours denote the carnitine shuttle (dark brown), the enzymes in the β-oxidation (white) and model extensions (purple) compared to [20] which were implemented in the last part of this paper. CPT1: carnitine palmitoyltransferase 1, CACT: carnitine acylcarnitine translocase, CPT2: carnitine palmitoyltransferase 2, SCAD: short-chain acyl-CoA dehydrogenase, MCAD: medium-chain acyl-CoA dehydrogenase, LCAD: long-chain acyl-CoA dehydrogenase (not present in human mFAO), VLCAD: very-long-chain acyl-CoA dehydrogenase, CROT: crotonase, M/SCHAD: medium/short-chain hydroxyacyl-CoA dehydrogenase, MCKAT: medium-chain ketoacyl-CoA thiolase, MTP: mitochondrial trifunctional protein, ETF: electron transfer flavoprotein, ETC: electron transport chain, ACOT: acyl-CoA thioesterases, PDH: pruyvate dehydrognase, SCFAs: short-chain fatty acids. **Conserved moiety**: total concentration of all CoA species in the matrix. **Boundary conditions:** Matrix concentrations of NADH, NAD⁺, FADH₂, FAD⁺, acetyl-CoA, free carnitine and cytosolic concentrations of free CoA, free carnitine, malonyl-CoA and palmitoyl-CoA were kept constant.

in the kinetic equations. This model is based on reversible, saturable enzyme-kinetic equations with competitive inhibition by alternative substrates and products. It contains experimentally determined parameters of rat-liver enzymes. Bistability has not been described yet for this model, but the competition between alternative substrates and products was found to give rise to another interesting phenomenon [2,9]. An increasing supply of palmitoyl-CoA (an acyl-CoA with 16 carbon atoms in acyl chain) led to a steep and detrimental decline of the mFAO flux, accumulation of intermediate acyl-CoA esters and depletion of the free CoA. The increased supply of palmitoyl-CoA mimics the elevated fatty acid levels in plasma of obese subjects [20,39] or during fasting [40]. Particularly the last enzyme in the pathway, the medium-chain ketoacyl-CoA thiolase (MCKAT), played a key role in the observed flux decline [37]. At increasing palmitoyl-CoA concentrations, the conversion of C6- and C4-ketoacyl-CoA was limited by inhibitory concentrations of upstream acyl-CoAs, effectively establishing a feedforward inhibition loop. In addition, there was an indirect substrate inhibition due to the conservation of the total CoA pool. At increasing palmitoyl-CoA more and more CoA was

sequestered by pathway intermediates. This reduced the availability of free CoA for MCKAT. The reduced MCKAT activity thus sparked a vicious cycle and severe jamming of the flux, analogous to [41]. Since the mechanism underlying the steep flux decline in the mFAO is so similar to the mechanism leading to bistability in dual MAPK phosphorylation [14], we revisit this issue here.

In this study we aimed to analyse the coexistence of multiple steady states in the mitochondrial fatty-acid oxidation in the context of health and disease. It is known that bistability does not only depend on the regulatory structure of the pathway, but also on the kinetic parameters [42]. Indeed, we will show that the above-described mFAO model [20] has a specific region in the parameter space in which two distinct stable and one unstable steady state can be attained with different fluxes. We traced the origin of the bistability back to the distributive kinetics and the conservation of CoA, in particular in the last rounds of β-oxidation when fatty acids have already been degraded to a length of four to six carbon atoms. We then extended the the mFAO model to investigate various interventions that may confer health benefits by activating the pathway, including (i) activation of MCKAT via its endogenous regulator p46Shc [33,43], (ii) addition of a thioesterase (an acyl-CoA hydrolysing enzyme) [44], and (iii) concomitant activation of a number of upstream and downstream enzymes by short-chain fatty-acids, metabolites that are produced from nutritional fibers in the gut [45]. A better understanding of the switch behavior of the mFAO between a low- and a high-flux state may lead to dietary and pharmacological intervention in the treatment or prevention of obesity and other metabolic diseases.

## Results

### Bistability and hysteresis in the mFAO pathway

To explore the stability behavior of the mFAO pathway, we performed a numerical study of the previously published mFAO model [20], without the extensions displayed in Fig 1. The [NAD$^+$]: [NADH] ratio was taken equal to 20, which is on the higher end of the measured range [46–49] and all other parameters were unchanged. We varied the supply of palmitoyl-CoA in the forward and reverse directions. At each palmitoyl-CoA concentrations, simulation started from the previous steady-state concentrations as a starting point. Three steady states were obtained between 104.8 and 136.4 μM of palmitoyl-CoA, and one steady state outside this range (Fig 2A). To evaluate the stability of the steady states, we computed the eigenvalues of the Jacobian matrix corresponding to each steady state (Fig 2B). In the low or high palmitoyl-CoA range the single steady state had a negative eigenvalue, indicating that it was stable. Between the critical points at which the system made the transition from one to three steady states, one eigenvalue had a positive real part, indicating that it was unstable. The eigenvalues of the other two steady states had negative real parts, indicating that they were stable. Thus, the mFAO model exhibited bistable behavior with the bistability region bounded by the critical palmitoyl-CoA concentrations 104.8 and 136.4 μM. In Fig 2A, within the bistability region, the two stable steady state solutions are displayed in red and dark blue and the unstable steady state solution in green. In the following, the unstable state will not be displayed anymore, for the purpose of visual clarity.

From Fig 2A it can be seen that the bistable mFAO pathway exhibited hysteresis behaviour. If the palmitoyl-CoA concentration was varied from low to high (red arrows) the pathway entered the bistable region at the high-flux steady state and followed the high-flux curve until it reached the critical 'switch-down' point, at 136.4μM, and jumped to the low-flux state. In contrast, if the mFAO started from a high palmitoyl-CoA concentration, it remained at the low-flux state as the palmitoyl-CoA concentration decreased (blue arrows). Thus, it followed

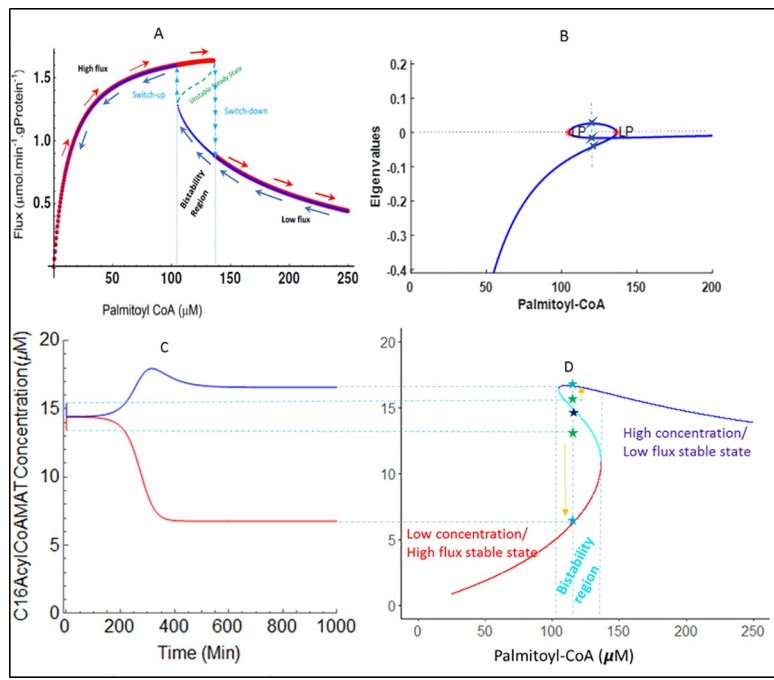

**Fig 2. Bistability and hysteresis in mitochondrial fatty acid oxidation.** The simulated steady state of mFAO uptake fluxes for increasing and decreasing palmitoyl-CoA concentrations are based on the rat liver model [20] with the [NAD+]: [NADH] ratio equal to 20 and all other parameters unchanged. A. Steady state uptake flux of palmitoyl-CoA for increasing (red curve) and decreasing (blue curve) palmitoyl-CoA concentrations. The arrows indicate in which direction the palmitoyl-CoA concentration was varied. B. The real parts of the eigenvalues of the Jacobian matrix plotted against the palmitoyl-CoA concentration. The bistable region between the critical points is characterized by two stable steady states, as can be inferred from their eigenvalues. C. Time course solution until convergence to a stable state starting from initial states close to the unstable stable. Cytosolic palmitoyl-CoA 120 μM, initial mitochondrial palmitoyl-CoA 13.41 (red) or 15.41 $\mu$M (blue). D. Steady-state mitochondrial against cytosolic palmitoyl-CoA; the light-blue line denotes the unstable state, and red and dark-blue lines, respectively the low- and high-flux state.

the low-flux curve through the bistability region, until it reached the 'switch-up' palmitoyl-CoA concentration at 104.8 μM and jumped to the high-flux state. A further decrease in palmitoyl-CoA then moved the mFAO flux down along the high flux steady state curve.

Close to the unstable steady state, the time course depended very sensitively to the initial metabolite concentrations. When, for instance, the initial mitochondrial palmitoyl-CoA concentration was chosen just above its unstable steady-state level, it would increase to reach the stable steady state at high mitochondrial palmitoyl-CoA (which corresponds to the low flux). If, in contrast it was chosen just below its unstable steady-state level, it would decrease to its low-concentration, high-flux state (Fig 2C and 2D).

## Analytical steady-state analysis of a reduced model

To understand the origin of the bistability, we would ideally obtain an analytical solution. For the entire mFAO model, this is currently not feasible. Therefore, we compressed the model into a one-dimensional ODE system. We considered the last part of the mFAO pathway (Fig 3, details in Materials and Methods section), which is involved in the production and consumption of ketoacyl-CoA with a length of four carbon atoms (C4-ketoacyl-CoA). This reduced pathway consisted of two enzymes (MSCHAD and MCKAT), and apart from C4-ketoacyl-CoA itself, also the preceding metabolite C4-hydroxyacyl-CoA and the final product acetyl-CoA. Solving the steady state equations for C4-ketoacyl-CoA (S2 Text) resulted in a cubic

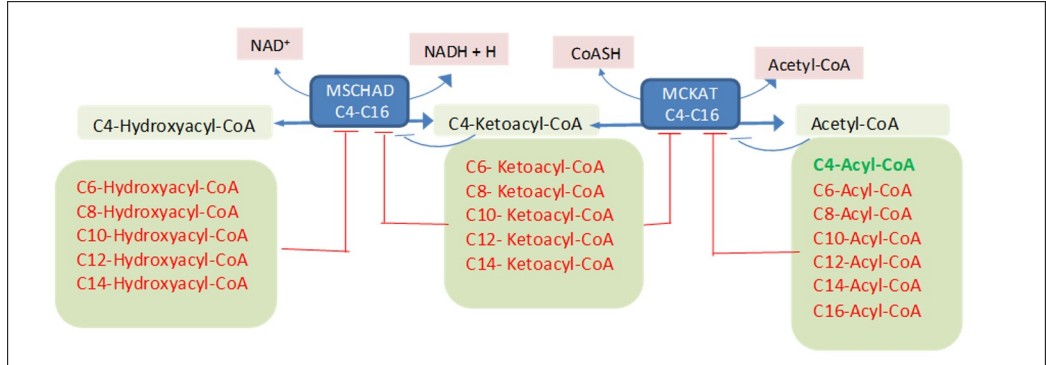

**Fig 3. Modular description of the mFAO pathway for one dimensional ODE model.** The module includes the conversion to and from C4-Ketoacyl-CoA using MSCHAD and MCKAT enzymes and competitive inhibitory products and substrates of other carbon chain length.

equation:

$$aM^3 + bM^2 + cM + d = 0 \tag{1}$$

in which M refers to the concentration of C4-ketoacyl-CoA. Depending on the parameters, a cubic equation has one or three solutions. Biologically speaking, this means that this simplified mFAO system has, depending on the conditions, one or three steady states. This demonstrates that the reduced pathway can in principle exhibit bistability.

## Origin and causes of mFAO bistability

Following Markevich at al. [14] we subsequently analysed the origin and possible causes of bistability of the mFAO pathway numerically in the complete model, based on (i) the feedforward inhibition resulting from the distributive kinetics and (ii) the moiety conservation of coenzyme A. To this end, we analysed the model in a modular way, focusing on a supply module that produces the intermediate metabolite C4-ketoacyl-CoA and a demand module consuming this compound. In this modular approach the rate of the C4-MSCHAD reaction represents the supply flux and the rate of the C4-MCKAT reaction the demand flux (Fig 4A). A motivation to modularise around C4-ketoacyl-CoA is that the demand reaction C4-MCKAT is inhibited by the upstream acyl-CoAs and therefore subject to direct inhibition by an upstream substrate. Moreover, this reaction requires free CoA and is therefore limited by sequestration of CoA when the acyl-CoAs accumulate. This is the indirect substrate inhibition, via moiety conservation. Finally, our previous analysis [37] showed that particularly the short-chain (C4 and C6) MCKAT reaction controlled the flux in the region of interest. Hence, we assume higher carbon chain length (C6-C16) substrate and product inhibition on this module are minimal or negligible noting the main role of C4-acyl-CoA (S1 Fig) in this module and the entire pathway.

Fig 4B shows the analysis of the modularised system at a fixed concentration of 120 μM palmitoyl-CoA, i.e. within the bistability region. The concentration of the linking metabolite C4-ketoacyl-CoA was kept constant over time, but the model was analysed for different concentrations of this metabolite. This reduced the model by one differential equation, and the rest of the differential equations were solved as before. The flux through the supply module (measured as C4-MSCHAD) was inhibited by an increasing concentration of its product C4-ketoacyl-CoA (Fig 4B). As a consequence (S2 Fig) the upstream C4-acyl-CoA accumulated. This phenomenon was amplified by the unfavourable equilibrium constant of C4-MSCHAD,

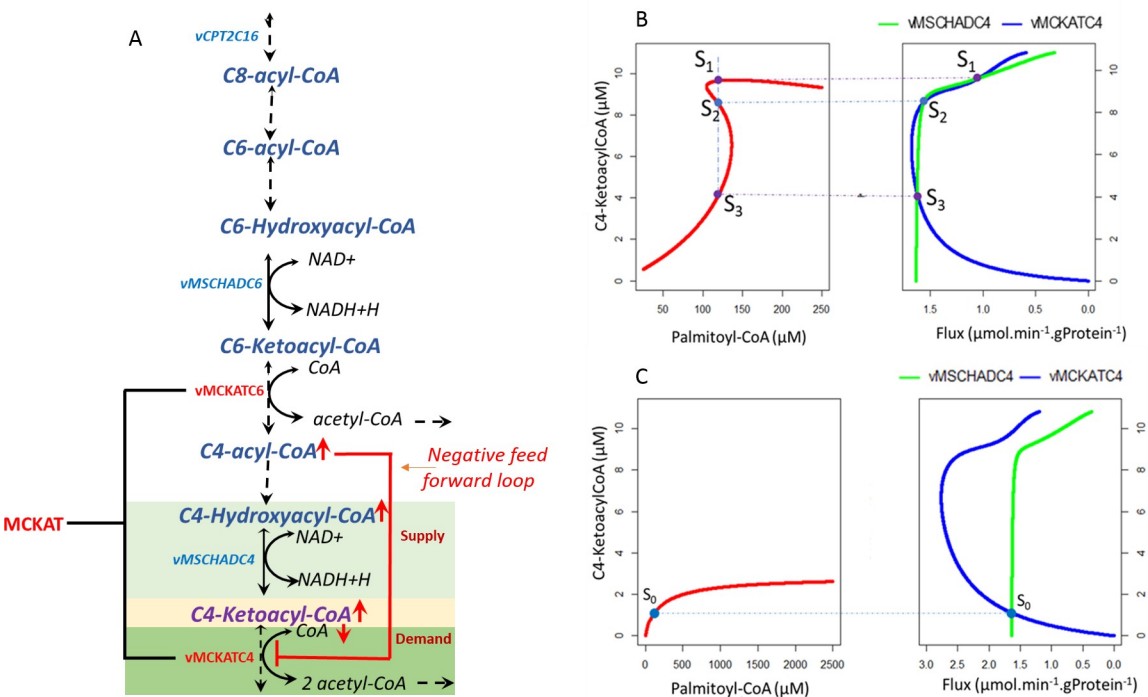

**Fig 4. Origin of mFAO bistability.** A. Modularisation of the mFAO pathway around C4-ketoacyl-CoA into a supply and a demand module. The flux of the supply and the demand module were plotted as a function of a fixed C4-ketoacyl-CoA concentration (B,C). B. Bistability of mFAO arises due to the presence of feedforward substrate inhibition (negative feedforward loop) of C4-acyl-CoA on vMCKATC4 (reaction rate of MCKAT for its C4 substrate) as shown by three steady states of mFAO (S1, S2, and S3) that correspond to three points of intersection between the rates vMCKATC4 and vMSCHADC4 (reaction rate of MSCHAD for its C4 substrate) around C4-ketoacyl-CoA. C. No bistability, but a single steady state as a result of removing the feedforward substrate inhibition (negative feedforward loop) of C4-acyl-CoA from vMCKATC4, corresponding to one point of intersection between the rates vMCKATC4 and vMSCHADC4. All calculations were done at a fixed concentration of 120 μM palmitoyl-CoA.

which favours the reverse reaction ($K_{eqmschad}$ = 0.000124). The accumulation of the C4-acyl-CoA and other CoA esters led to a sequestration of CoA and therefore a decline of free CoA (S2 Fig). The flux through the demand module (C4-MCKAT) requires its substrate C4-ketoacyl-CoA, hence started at zero flux and increased initially with increasing C4-ketoacyl-CoA (Fig 4B). However, the accumulation of C4-acyl-CoA and the decline of CoA inhibited the C4-MCKAT flux when C4-ketoacyl-CoA was increased further (Fig 4B). Altogether this yielded the three points of intersection between supply and demand of C4-acyl-CoA (Fig 4B). These points correspond to the three distinct steady states (S1, S2, and S3 in Fig 4B) that we also found via eigenvalue analysis. The steady states S1 and S3, which correspond to high and low concentrations of C4-ketoacyl-CoA, were stable, whereas the intermediate state S2 was unstable. This can be seen from Fig 4B: outside the stable points the rate of production of C4-ketoacyl-CoA (green line, $v_{MSCHADC4}$) differs from its rate of consumption (blue line, $v_{MCKATC4}$). If the production is faster than the consumption, the concentration of C4-ketoacyl-CoA will increase, and vice versa. Thus, it can be inferred that the system tends to move towards steady states S1 or S3 and away from S2. This is consistent with the previous eigenvalue analysis (Fig 2).

To further verify the role of the substrate inhibition, we artificially removed the inhibition of C4-MCKAT by C4-acyl-CoA (Eq 4, see Materials and Methods for details). Indeed, in the absence of the feedforward substrate inhibition by C4-acyl-CoA, the rates of C4-MCKAT and

C4-MSCHAD yielded only one point of intersection ($S_0$ in Fig 4C), corresponding to a single steady state. Interestingly, the C4-MCKAT flux still decreased at high C4-ketoacyl-CoA, due to the sequestration of CoA (S2 Fig), but this inhibition was too weak to elicit extra intersection points between the supply and demand fluxes. Thus, these results suggest that the kinetics of the last cycle of β-oxidation, with substrate inhibition of the C4-MCKAT reaction by C4-acyl-CoA in combination with moiety conservation of CoA, play a pivotal role in the origin of the bistability.

## Sensitivity of mFAO bistability to physiological conditions

Physiologically, we consider the high-flux state as a healthy state. The mFAO pathway is critical for the supply of energy during fasting, to support growth or exercise. Moreover, a low mFAO capacity may lead to accumulation of fat and thereby to obesity or type 2 diabetes. Pharmacological or physiological interventions to stimulate the mFAO pathway have long been a topic of intense research [50–54]. Interventions that shift the pathway from a low to a high-flux state or remove the low-flux state altogether, may therefore be promising for therapy. In this section we analyse how various cues affect the bistability behaviour of the pathway.

[NAD$^+$]:[NADH] ratio. The [NAD$^+$]:[NADH] ratio is a measure of the redox state of cells or cellular organelles. In the mFAO pathway NAD$^+$ is a co-substrate for the reactions catalysed by MSCHAD and MTP (Fig 1). Previously, we found with the same computational model that the [NAD$^+$]:[NADH] ratio in the mitochondrion was among the top 4 parameters with highest response coefficients and thus the highest impact on the mFAO flux [37]. Moreover, experimental evidence suggests that the values of [NAD$^+$]:[NADH] ratio in different cell compartments control cell metabolism [55,56]. Therefore, we examined the effect of varying the [NAD$^+$]:[NADH] ratio on the bistability and hysteresis behaviour of the mFAO flux. Increasing the [NAD$^+$]:[NADH] ratio revealed a shift in the critical palmitoyl-CoA concentrations and the bistable region of mFAO flux (Fig 5A). For a small value of the ratio (e.g. 15) the bistability region exists, but was barely visible, since it covered only a very small range (dark yellow curve in Fig 5A) (S6 Fig displays magnified small bistability regions). For a smaller ratio (e.g. 10 or 12) the flux declined but with no bistability (red and orange curves in Fig 5A). In contrast, for larger ratios (e.g. 20 or 23, cyan and blue curves) the bistability region covered a wider range of palmitoyl-CoA concentrations and also a wider flux range. However, at a very high [NAD$^+$]:[NADH] ratio (40 or above), the bistability disappeared, the low-flux state disappeared, and the pathway exhibited saturation kinetics, at least up to 250 μM of palmitoyl-CoA (magenta curve). If instead the total pool of [NAD$^+$] plus [NADH] was varied between 100 and 300 μM, whilst keeping the [NAD$^+$]:[NADH] ratio at 20, bistability existed over the range between 200 and 275 μM (S7 Fig).

[FAD]:[FADH$_2$] ratio. FAD is another redox co-factor, which in contrast to NAD$^+$, is bound to proteins. In the mFAO pathway, FAD is reduced to FADH$_2$ by the acyl-CoA dehydrogenases (SCAD, MCAD, LCAD, and VLCAD in Fig 1). Also the [FAD]:[FADH$_2$] ratio was previously found to have a large impact on the flux [37]. Bistability existed for a wide range of [FAD]:[FADH$_2$] ratios (Fig 5B)). Increasing the [FAD]:[FADH$_2$] ratio from 0.10 to 0.93 shifted the critical values and the bistable region to higher palmitoyl-CoA concentrations (Fig 5B). For high [FAD]:[FADH$_2$] ratios (e.g. 0.79, 0.67, 0.54, 0.48) the bistability region covered a relatively wide to small ranges of palmitoyl-CoA (bounded by blue, cyan, green and dark yellow curves, respectively) (S6 Fig displays magnified small bistability regions). In contrast, for small [FAD]:[FADH$_2$] ratios (e.g. 0.10, 0.28 or 0.40) the system exhibited no bistability (dark red, red and orange curves). At a very high ratio of [FAD]:[FADH$_2$] (0.93 or above), the mFAO bistable behavior disappeared and the mFAO exhibited saturation kinetics (magenta

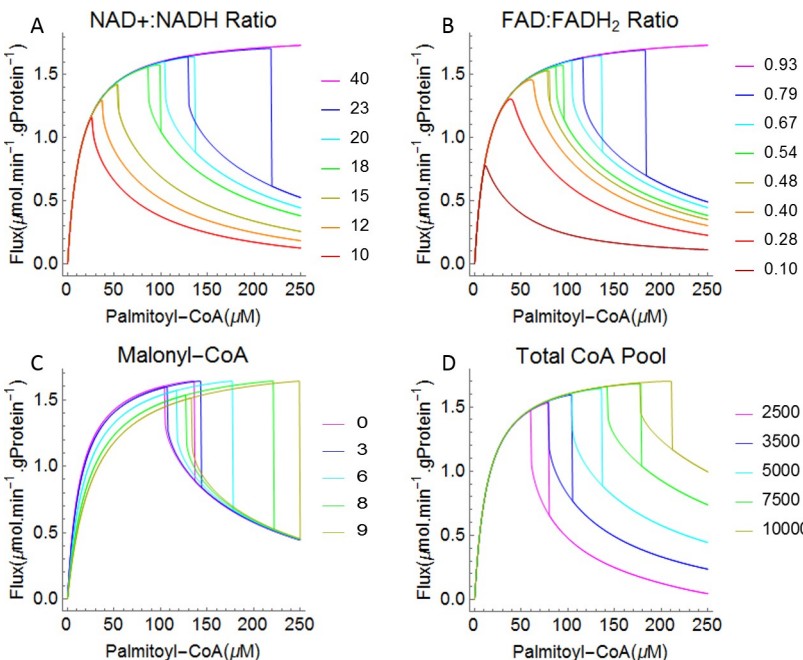

**Fig 5. Bistability and hysteresis behaviour in mitochondrial fatty acid oxidation when varying selected model parameters.** Simulated uptake fluxes for increasing and decreasing palmitoyl-CoA concentrations. A. [NAD$^+$]:[NADH] ratio varying between 10 and 40, without altering the sum of [NAD$^+$] and [NADH]. B. [FAD]:[FADH2] ratio varying between 0.10 and 0.93. C. Malonyl-CoA varying between 0 and 10 μM. D. Total CoA varying between 2500 and 10000 μM.

curve), up to 250 μM of palmitoyl-CoA. Overall, the bistability responded similarly to shifts in the [FAD]:[FADH$_2$] or the [NAD$^+$]:[NADH] ratio.

**Malonyl-CoA.** Malonyl-CoA, a precursor for fatty acid synthesis, is a well-known competitive inhibitor of CPT1 [57], the first enzyme of the mFAO pathway, involved in transporting long chain fatty acids into the mitochondria (Fig 1). So far all results presented were obtained in the absence of malonyl-CoA. Malonyl-CoA was varied between 0 and 10 μM, which is close to the $K_i$ of CPT1 for malonyl-CoA of 9.1 μM [20]. As expected, addition of increasing concentrations of malonyl-CoA inhibited the flux. In addition, however, it increased the range of the bistability region substantially (Fig 5C). The maximum flux was the same for all malonyl-CoA concentrations. However, at higher malonyl-CoA concentrations the maximum flux was reached at higher palmitoyl-CoA concentrations. This shows that an increase of the substrate for CPT1 can compensate for the inhibition of the enzyme by malonyl-CoA. Similarly, the flux of the low-flux steady state also became independent of malonyl-CoA at higher palmitoyl-CoA concentrations. Physiologically, it is interesting that the high-flux branch existed until much higher concentrations of palmitoyl-CoA when malonyl-CoA was added. Due to the stretching of the high-flux branch, malonyl-CoA can therefore activate the mFAO flux even though it acts through inhibition of one of its enzymes. Analogously, if the $V_{max}$ of CPT1 was decreased, bistability also set in at a higher palmitoyl-CoA concentration (S8 Fig).

**Effect of total CoA.** In the computational model, the total concentration of CoA in the mitochondria (sum of free CoA and CoA esters) was fixed at 5 mM. In reality, the biosynthesis of CoA responds to feeding-fasting cycles [58]. The free CoA concentration has been identified as an pivotal metabolite to explain the flux decline in the mFAO model [37]. We wondered if

the total CoA concentration may also affect the bistability properties of the mFAO pathway. Indeed, as the concentration of total CoA increased, the bistability region shifted to higher concentrations of palmitoyl-CoA (Fig 5D). Similarly, as the concentration of total CoA increased, the accumulation of acyl-CoAs and depletion of free CoA shifted to higher concentrations of palmitoyl-CoA (S5 Fig). The maximum flux did not depend on the total CoA concentration, but the low-flux steady state was much higher at elevated total CoA concentrations (Fig 5D).

## Flux control analysis and bistability

Martines et al. [37] performed a metabolic control analysis (MCA) for increasing levels of palmitoyl-CoA, but without considering bistability. Here we revisited the MCA and distinguished between the low-flux and high-flux solutions. The flux control coefficient $C_i^J$ [59,60] of an enzyme i over the steady-state flux J is defined as:

$$C_i^J = \frac{dlnJ/dp}{\partial \ln v_i / \partial p} \qquad (2)$$

in which p is a parameter that only affects the rate $v_i$ of enzyme i. Numerically, the flux control coefficients were computed by a small (0.0001%) change of the maximal velocity Vmax of the enzymes and a re-evaluation of the steady-state flux. Thus, the flux control coefficients expressed the extent to which the steady-state flux responded to an activation of a specific enzyme.

Distinct flux control coefficients were computed for increasing and decreasing levels of palmitoyl-CoA, respectively (Fig 6 (right and left panels)). At palmitoyl-CoA concentrations below the bistability region and at the high-flux state within the bistability region, CPT1 had a flux control coefficient of 1, implying that it fully controlled the flux. A flux control coefficient of 1 means that the steady state flux changes proportionally to an activation of the enzyme. In accordance with the summation theorem for flux control coefficients [59,60], the flux control coefficients of all other enzymes were negligible under these conditions, such that their sum remained 1. In contrast, at palmitoyl-CoA concentrations beyond the bistability region and at

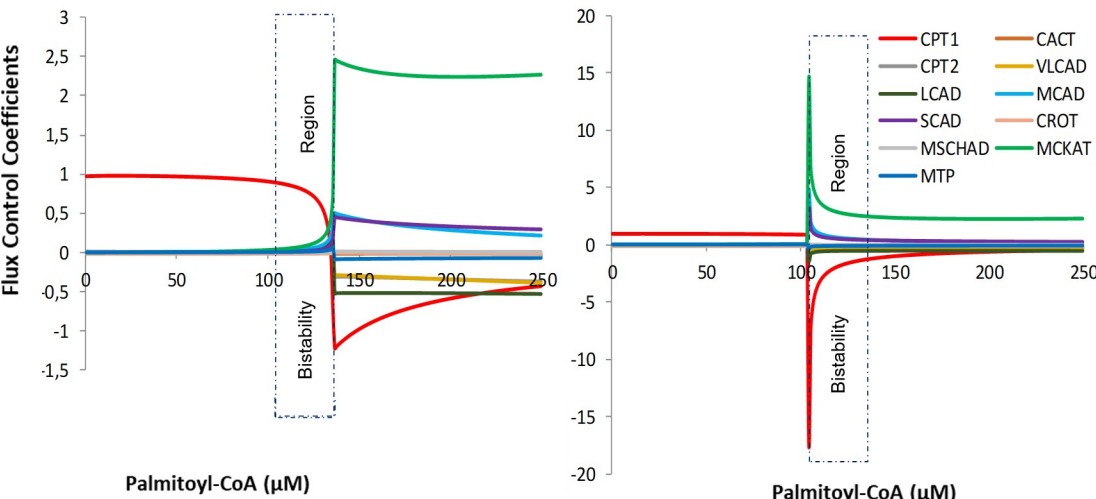

**Fig 6. Flux control coefficients and bistablity in mFAO model.** Flux control coefficients of 11 enzymes for (right) increasing supply of palmitoyl-CoA concentrations and (left) decreasing supply of palmitoyl-CoA concentrations.

the low-flux state within the bistability region, CPT1 exerted a negative control over the flux. This was compensated by high positive control exerted by MCKAT, and to a lesser extent by MCAD and SCAD (Fig 6 right panels). Qualitatively, these results agree to those of Martines et al [37], the key difference being that we used a higher [NAD$^+$]:[NADH] ratio to enlarge the bistability region. Moreover, our results show that the distribution of flux control in the high-flux state follows that of the low palmitoyl-CoA concentration where the flux ascends with an increasing substrate concentration. In contrast, in the low-flux state the distribution of flux control connects to that at high substrate concentrations. The highly negative flux control exerted by CPT1 in the low-flux state means that an activation of the enzyme causes a steep flux decline. This is not surprising, since it has the same effect as a higher palmitoyl-CoA concentration, namely to funnel more substrate into the pathway and aggravate the accumulation of intermediate metabolites and sequestration of CoA [20,37].

## mFAO regulation and bistability

Based on the results of this study and elsewhere [20,37] we conclude that bistability and the decline of the mFAO flux upon substrate overload are intrinsic properties of the isolated mFAO model, if it is not regulated adequately. In reality, however, the pathway is embedded in a larger network of reactions, which may attenuate or aggravate this behaviour. To study the role of the surrounding network, we simulated three mechanisms that may have evolved to protect the pathway against the low-flux state.

**Regulation of MCKAT by p46Shc protein.** In the low-flux state, MCKAT exerted the highest positive control over the flux of all mFAO enzymes. Endogenously, the MCKAT enzyme is inhibited by the p46Shc protein [33]. p46Shc is a splice variant of Shc, which is imported into the mitochondria where it binds specifically to MCKAT. Ablation of p46Shc was shown to activate the mFAO flux and prevent obesity in mice [33]. In particular, a 30% increase in palmitate oxidation capacity was observed for purified mitochondria from ShcKO (20% residual activity of p46Shc protein) mice compared to controls (WT levels of p46Shc). The Shc protein is an interesting target for pharmacological intervention. Inhibition of another splice variant, p52Shc, was shown to sensitize mice towards insulin. The catalytic capacity of MCKAT was made a function of p46Shc activity according to equation E1 (S1 Text). Decreasing p46Shc protein activity, thus activating MCKAT, stimulated the mFAO flux specifically in the low-flux state, where MCKAT had a high flux control coefficient (Fig 6). Interestingly, reduced p46Shc protein activity removed or shifted the entire bistability region (red curve for WT) to higher palmitoyl-CoA concentrations (Fig 7A). The flux in the high-flux state was not affected by p46Shc. In the ShcKO, the flux followed saturation kinetics, avoiding the flux decline and bistability at least up to 250 μM palmitoyl-CoA (blue curve in Fig 7A).

**Acyl-CoA thioesterases as a safety valve.** Conversion of free fatty acids to acyl-CoAs is necessary for their entry into fatty acid metabolism [61]. However, excessive accumulation of intracellular acyl-CoAs can limit the mFAO flux due to CoA depletion. Interestingly, acyl-CoA thioesterases (ACOT) that catalyse acyl-CoA hydrolysis to free fatty acids and release CoA, can facilitate mFAO in the liver and other organs [62–64]. This is not trivial, however, since ACOT competes with the mFAO for acyl-CoAs (Fig 1). We expect, therefore, that the mFAO flux is only increased by ACOT activity if CoA becomes limiting, i.e. in the low-flux state. Until here all results were obtained with zero ACOT activity. We tested if and when ACOT could rescue the decline of the mFAO flux. Until now we plotted the palmitoyl-CoA consumption flux as a measure of the mFAO flux. Since a steady state could only be reached if all palmitoyl-CoA was fully oxidized to acetyl-CoA with the concomitant production of NADH and FADH2 (Fig 1)), the relative results would have been identical if we had plotted

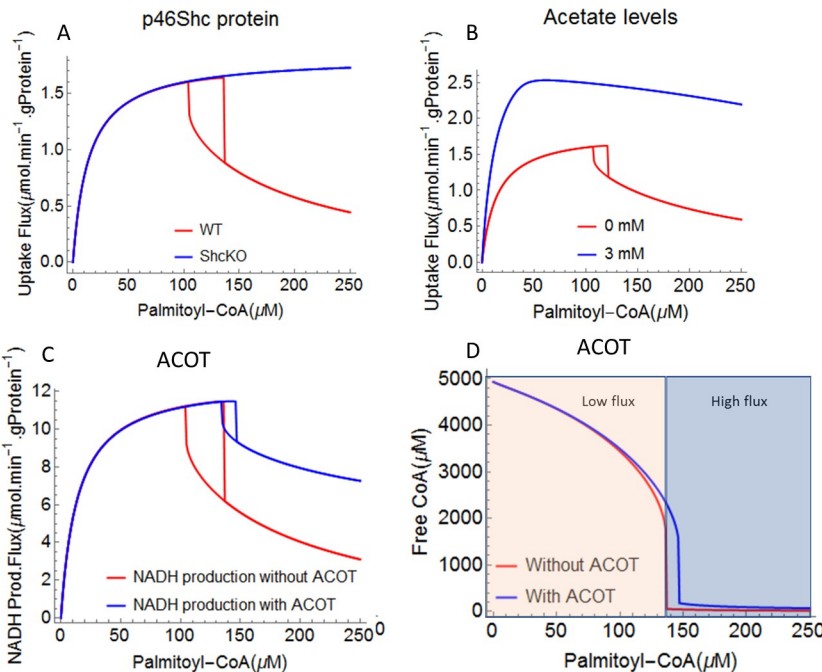

**Fig 7. Model extensions and mFAO bistability.** The computational model was extended with the different reactions indicated in Fig 1 and detailed in S1 Text. A. Activation of MCKAT via reducing the inhibitory effect of p46Shc protein (WT: wild type and ShcKO: 20% residual activity of p46Shc protein. B. The model was extended with PDH and ETC and activation of CPT1 and ETC by SCFA. C and D. The model was extended with ACOT.

the production of NADH. Only the absolute fluxes would have differed by a factor of 7, since palmitoyl-CoA undergoes 7 rounds of mFAO. After the model was extended with ACOT (S1 Text), however, a fraction of the consumed palmitoyl-CoA was rerouted into ACOT and not oxidized (S4 Fig). Since we were interested in the full oxidation of fatty acids, we here plot the NADH production. Indeed the low-flux state was strongly elevated by the addition of ACOT, without an effect on the high-flux state (Fig 7C). The model with ACOT activity (blue curves) demonstrates a shift in the bistability region to higher palmitoyl-CoA concentrations compared to the model without ACOT (red curves) (Fig 7C). Interestingly, this effect is reached at a low ACOT flux (below 8.6% of the palmitoyl-CoA consumption flux, whereas the downstream NADH production flux is increased by up to 31% (S3 Fig). This implies that the loss of ATP in the acyl-CoA synthetase / ACOT vicious cycle is more than compensated by the increased production of NADH and $FADH_2$, which feed into the respiratory chain to produce ATP. As expected, the free CoA concentration was elevated substantially by ACOT, specifically in the low-flux state (Fig 7D). The efficiency of NADH produced to palmitoyl-CoA produced was somewhat reduced by ACOT, but this happened merely in the low-flux state (S4 Fig).

**Regulation by short chain fatty acids (SCFA).** SCFAs, such as acetate, butyrate, and propionate, are produced by microbial fermentation of dietary fiber in the gut. SCFAs are increasingly appreciated as pivotal mediators that link diet and gut microbiota to host physiology by modulating the activity of enzymes and transcription factors [45,65]. SCFAs, particularly acetate and butyrate protect against obesity and type 2 diabetes in humans [66–69]. Studies in mice demonstrated that SCFA supplementation protects against obesity on a high-fat diet [70,71]. SCFA activated the mFAO directly by (i) increasing the activity of CPT1 by two fold [70], (ii) decreasing the production of the CPT1 inhibitor malonyl-CoA, and (iii) increasing the demand for NADH via uncoupling of the respiratory chain by activation of UCP2 [70].

UCP2 increases the [NAD$^+$]:[NADH] ratio by stimulating the oxidation of NADH to NAD+ [72,73].

In order to get insight into the role of the SCFA concentrations on the bistability of the mFAO, the model was extended with pyruvate dehydrogenase (PDH) and an electron-transport chain (ETC), and the catalytic capacities of CPT1 and the ETC were made a function of the acetate concentration according to equation E2 (S1 Text) [70]. Effectively, this made the [NAD$^+$]:[NADH] ratio a variable instead of a fixed boundary parameter. Without acetate or at a concentration up to 0.5 mM, bistability was observed, but it disappeared at higher concentrations (Figs 7B and S9)). Interestingly, acetate increased the mFAO flux both in the low-flux and in the high-flux state (Figs 7B and S9).This can be understood from the fact that both CPT1 and NADH oxidation were stimulated by acetate. CPT1 exerts most flux control in the high-flux state, while downstream reactions including the [NAD$^+$]:[NADH] ratio take over control in the low-flux state (Fig 6 and [37]).

## Discussion

In this paper we demonstrated that the mitochondrial oxidation of fatty acids can exhibit bistability and hysteresis with increasing and decreasing supply of long chain fatty acid substrates. In the bistable region, a stable steady state was found with either high or low flux, depending on the previous state of the fatty acid oxidation. Outside the bistability region, only one possible steady state flux existed. Excessive accumulation of fat in body tissues is a risk factor for multiple diseases. Tissue fat content is the resultant of uptake, oxidation and *de novo* synthesis. Our computational model was constructed for fatty acid oxidation in the liver. Therefore, the results have a direct impact on fat accumulation in the liver, such as in non-alcoholic fatty liver disease (NAFLD) [74].

Using the same computational model of mitochondrial fatty acid oxidation, we previously showed that the flux declined steeply with increasing substrate concentrations [20]. We then found that removing the competition between different substrates and products for binding to the same enzymes precluded the flux decline. Hence, it was clear that the catalytic capacity of the enzymes was sufficient to support a high flux state. Here we found that, in specific regions of the parameter space, these two states can even coexist. Which state is reached depends on the history of the system, in particular whether it comes from a high or a low palmitoyl-CoA concentration.

To identify the mechanism leading to bistability, we focused on the similarities between the fatty-acid oxidation and MAPK signalling [14]. In both pathways the kinetics are distributive, i.e. intermediates in different rounds of catalysis are converted by the same enzyme(s). Moreover, in both cases moiety conservation is involved, either conservation of CoA (this study) or of the MAPK protein that undergoes phosphorylation (in the signalling study). We therefore adopted the same supply-demand approach that had been used to analyse bistability in MAPK signalling [14]. We focused on the last cycle of mitochondrial β-oxidation, i.e. from C4-Acyl-CoA to two molecules of acetyl-CoA, and particularly on the last reaction of the cycle, C4-MCKAT. From previous analyses of the same computational model [20,37] it was known that particularly the short-chain C4- and C6-acyl-CoA tend to accumulate upon substrate overload. In this system negative feedforward arises from two different mechanisms. First, the flux through C4-MCKAT requires the co-substrate CoA. When the pathway is overloaded with substrate, the co-substrate CoA is sequestered in CoA esters, among which C4-acyl-CoA makes a large contribution. As a result, an increase in C4-acyl-CoA results in a decrease in CoA that leads to apparent substrate inhibition. Second, C4-acyl-CoA is the substrate for the last cycle of β-oxidation and thereby supplies MCKAT with its direct substrate C4-ketoacyl-

CoA. It is at the same time, however, the product of the previous round of β-oxidation and thereby an inhibitor of MCKAT. Also this mechanism leads to apparent substrate inhibition or negative feedforward [13]. Our findings require further experimental validation analogous to bistability in glycolysis [16].

Another mechanism of bistability in liver fat deposition that has been described by a computational model, concerns the balance between fat uptake, oxidation and storage [75]. In this model the swelling of hepatocytes by fat deposition is considered to limit the flow of oxygen into the tissue. Thus, an increased supply of triglycerides will lead to lower availability of oxygen for fatty-acid oxidation and thus to apparent substrate inhibition of this pathway. Also this network was shown to exhibit bistability, with a state of low and a state of high triglyceride content. To the best of our knowledge, this mechanism has not been validated experimentally either. It is plausible, however, that both mechanisms reinforce each other. A limited availability of oxygen exerts its effect on the β-oxidation through reducing the ratio of [NAD$^+$]: [NADH], which we identified as a key parameter for the bistability within the β-oxidation pathway. Quantitative measurements of the [NAD$^+$]:[NADH] ratios in the mitochondrial matrix are based on the product/substrate ratio of malate dehydrogenase or β-hydroxybutyrate dehydrogenase, under the assumption of a certain pH and thermodynamic equilibrium [49]. We found bistability only on the higher end or even beyond the measured [NAD$^+$]:[NADH] range [47,48]. Although the model has been validated experimentally [20], we have to keep in mind, that all parameters may vary between conditions and have a certain error. Therefore, at this stage the qualitative conclusions–that bistability can occur and which parameters have an impact–is most relevant. It is plausible that the interaction between different parameters may shift the pathway in and out of the bistability region.

NAFLD is a the leading cause of chronic liver disease [74]. It is characterized by triglyceride accumulation in the liver and is strongly associated to obesity and metabolic syndrome. In a subset of NAFLD patients, the disease progresses into liver cirrhosis or hepatocellular carcinoma. The existence of bistability in liver fat metabolism may have an impact on potential treatments. In the region of bistability a transient perturbation or treatment would be sufficient to shift the liver from the unhealthy, low flux, high-fat state to the healthy, high-flux, low-fat state. Insight into the factors that govern the bistability region and the path between the healthy and the disease states are therefore of importance. Experimentally, there is sparse evidence for bistability in lipid handling. C57BL/6J mice on a high-fat diet show a large interindividual heterogeneity, with some mice becoming very obese, while others remain lean [76–79]. In one study [77] the differences between 'non-responders' (the mice that remain lean on a high-fat diet) and the 'strong responders' (those that become obese) were attributed in part to heterogeneity in the β-oxidation, based on urine metabolite profiles. In many studies, however, non-responders are excluded [76,79,80], thus precluding an in depth investigation of the underlying causes. Another condition in which bistability in lipid metabolism might play a role, is the sudden and dramatic occurrence of hypoketotic hypoglycemia in infants with a fatty-acid oxidation deficiency[21–24]. Obviously the production of ATP by oxidation of fatty acids is important for gluconeogenesis, while acetyl-CoA is a precursor for keton-body synthesis. The fact that the milder patients, e.g. those with an MCAD deficiency, show these symptoms only sporadically and without warning, suggests a threshold phenomenon. The possibility of bistability of lipid handling warrants revisiting both these issues.

The factors that promote the high-flux state in the fatty-acid oxidation or make the bistability vanish altogether in favour of the high-flux state, should be considered for therapeutic or dietary interventions. The key factors that we identified were the [NAD$^+$]:[NADH] ratio, the [FAD]:[FADH$_2$] ratio, the malonyl-CoA concentration, the concentration of the p46Shc protein, the CoA concentration, and finally the SCFAs. A higher NAD$^+$ concentration in the

hepatocytes promotes complete fatty acid oxidation [59]. The mitochondrial [NAD$^+$]:[NADH] ratio responds to the diet. For instance, during fasting or on a ketogenic diet, the mitochondrial [NAD$^+$]:[NADH] ratio in the liver tends to decrease [81]. The mitochondrial redox state can also be altered by metabolic diseases. Of interest, patients suffering from multiple acyl-CoA dehydrogenase deficiency disease have a partial deficiency in the electron transfer flavoprotein, which leads to a higher FADH$_2$ concentration and inhibits the oxidation of acyl-CoA esters [82]. At higher malonyl-CoA concentrations, we observed that the pathway maintained the high-flux state until higher palmitoyl-CoA concentrations, suggesting that the mild inhibition of CPT1 by malonyl-CoA could prevent a flux decline. Since malonyl-CoA is typically high in the fed state when fatty acids are low, one might wonder whether this is of physiological relevance. Possibly, such situations could occur when the liver is simultaneously overloaded with fat and sugars, particularly fructose, on a high-fat-fructose diet [83]. Moreover, the concentrations that we used (0–10 μM) are still in the low range compared to experimental values (2–60 μM, based on 0.4–15 nmol / g liver [84,85], 112 mgProtein/gram rat liver [86], and a cytosolic volume of $2.2 \cdot 10^{-6}$ L/mg protein [87]).

Among the key factors, the Shc protein may be most interesting for pharmacological interventions. Inhibition of p46Shc alleviates the inhibition of MCKAT, increases mFAO, and protects against obesity [33]. We predict that at low p46Shc activity the bistability vanishes and only a high-flux state with saturation kinetics remains. In vitro drug screening identified idebenone as a drug that binds with nM affinity to Shc. However, this compound binds most strongly to the p52 isoform of Shc, which regulates insulin sensitivity [88]. Caution should be taken, since p46Shc has a function in liver regeneration [89]. Furthermore, the addition of ACOT to the model stimulated the mFAO flux and shifted the bistability region to higher palmitoyl-CoA concentrations. This is explained by the fact that the low flux state is caused by accumulation of intermediary CoA esters and sequestration of free CoA [20,37]. ACOT relieves this condition. These modelling results are in line with experimental evidence that ACOT activity activates mitochondrial fatty acid oxidation in the liver [62]. Finally, it is known that SCFAs (acetate, propionate, butyrate) or the dietary fibers from which they are derived, stimulate the fatty-acid oxidation and protect against obesity and fatty liver [70,90–92]. Our simulations recapitulated this stimulation of the fatty-acid oxidation flux. More importantly, a high concentration of SCFAs also led to a disappearance of the bistability and just like in the case of Shc inhibition, only the high-flux state remained. The interventions that make the bistability disappear, are particularly interesting for short therapeutic interventions. Even a transient intervention would get the system back in the high flux state. For instance, when people take a high-fiber diet, the SCFA concentrations in the gut and the circulation are fluctuating between meals. In the case of a bistable system, however, the system would remain in the high flux-state after the intervention, since this is a stable state.

In conclusion, the possibility of bistability in the fatty-acid oxidation warrants different analysis of heterogeneity in physiological studies and leads to a new perspective for interventions in non-alcoholic fatty-liver disease and obesity.

## Materials and methods

### Computational model of mFAO pathway

A kinetic metabolic computational model of mFAO pathway (Fig 1) for rat liver was constructed in [20]. In our work, this computational model is first considered to investigate mFAO bistability and hysteresis behavior and secondly, we extend the model to include partial prevention of acyl-CoA esters accumulation and enhancing the activities of β-oxidation and oxidative phosphorylation via increasing the concentration of SCFAs. These extensions

provide remedial solutions to prevent the flux decline in the original model [20] at high fatty acid substrate concentration.

The model formulation considered the transport of palmitoyl-CoA from the cytosol to the mitochondrial matrix via the carnitine shuttle, the β-oxidation that involves various enzymes of different carbon length specificity, and a reversible conversion of C4-C16 acyl-CoAs into acyl-carnitines that can be transported between the matrix and the cytosol (Fig 1). Basically the model consists of two cellular compartments: the cytosol and the mitochondrion matrix, and their volumes are given along with initial concentrations of metabolites [20]. The original model of [20] consists of 45 ordinary differential equations (ODEs), 59 kinetic equations of reaction rates and 234 model parameters. Many of the parameter values in S1 Text and [20] were obtained from the literature that are determined from experimental data. In this work, all the model parameters remained the same except the NADH value changed from 16 μM to 12μM that was due to a change in the [NAD+]:[NADH] ratio from 15 to ~20 [46].

The ODE models of the mFAO pathway take the general form

$$\frac{d[C]}{dt} = G([C]), \text{ with } G([C]) = \sum_j \pm v_j([C]), \tag{3}$$

where $[C] = ([C_1], [C_2], \ldots, [C_N])^T$ a vector of concentrations of N metabolites and $G([C]) = (G_1([C_1]), G_2([C_2]), \ldots, G_N([C_N]))^T$ is functions of rate equations that accounts for the dynamics of N metabolites over time. The sign of reaction rates, $v_j[C_k]$, depends on whether an enzyme produces (+) or consumes (-) a given metabolite $C_k$. An example of such an ODE model, for the dynamics of C4-ketoacyl-CoAMAT concentration over time, is

$$\frac{d[C4KetoacylCoAMAT]}{dt} = \frac{vMSCHADC4 - vMCKATC4}{VMAT},$$

where VMAT the volume of the mitochondria and vMSCHADC4 and vMCKATC4 are rate equations for the conversion of C4-ketoacyl-CoA in the mitochondria by MSCHAD and MCKAT enzymes, respectively. In the model all reactions (e.g., vMSCHADC4 and vMCKATC4) were treated as reversible and product sensitive. Considering the competition for substrates, the rate equations $v_j([C])$ were mainly a modified reversible Michaelis-Menten equations with competitive inhibitions. An example of such a rate equation, vMSCHADC4 for the conversion of C4-ketoacyl-CoAMAT by the enzyme MCKAT, is given by:

$$vMCKATC4 = \frac{sf_{vmckatC4} V_{max,mckat} \left( \frac{[C4KetoacylCoA] \cdot [FreeCoA]}{Km_{MCKAT,C4KetoacylCoA} \cdot Km_{CoA}} - \frac{[AcetylCoA] \cdot [AcetylCoA]}{Km_{MCKAT,C4KetoacylCoA} \cdot Km_{CoA} \cdot K_{eq,MCKAT}} \right)}{\left( 1 + \frac{[C4AcylCoA]}{Km_{MCKAT,C4AcylCoA}} + \underbrace{\frac{[AcetylCoA]}{Km_{AcetylCoA}}}_{\substack{\text{inhibition by} \\ \text{reaction product}}} + \sum_{n=6}^{16} \underbrace{\left\{ \frac{[CnKetoacylCoA]}{Km_{MCKAT,CnKetoacylCoA}} - \frac{[CnAcylCoA]}{Km_{MCKAT,CnAcylCoA}} \right\}}_{\substack{\text{inhibition by substrates and products of} \\ \text{other chain length}}} \right) \left( 1 + \frac{[FreeCoA]}{Km_{CoA}} + \frac{[AcetylCoA]}{Km_{AcetylCoA}} \right)} \tag{4}$$

where the metabolites in the grey box are the substrates and products of competing reactions and act as inhibitors [20,37]. The parameter $sf_{vmckatC4}$ is a factor that is specific for the chain length of the substrate (here C4). Detailed description of the model are given in S1 Text in [20].

In the construction of this computational model, the total concentration of CoA has been considered as a conserved moiety, i.e. free CoA and CoA esters are only interconverted and CoA synthesis was not included. In addition, matrix concentrations of NADH, NAD+, FADH2, FAD+, acetyl-CoA, free carnitine and cytosolic concentrations of free CoA, free carnitine, malonyl-CoA and palmitoyl-CoA were kept constant.

## Steady state and bistability

The computational model is used to evaluate all the possible steady state solutions that correspond to a set of concentrations leading to $\frac{d[C]}{dt} = 0$. A pathway with bistable behavior is one with distinct two stable steady state solutions. To determine a given steady state is stable, the eigenvalues of the Jacobian matrix

$$J[C] = \begin{bmatrix} \frac{\partial G_1}{\partial C_1} & \dots & \frac{\partial G_1}{\partial C_N} \\ \vdots & \ddots & \vdots \\ \frac{\partial G_N}{\partial C_1} & \dots & \frac{\partial G_N}{\partial C_N} \end{bmatrix}$$

should have negative real part when evaluated at the steady state solution. Here, the path of steady state solutions for the concentrations of palmitoyl-CoA varying between 0 and 250 μM were obtained using NDSolve and FindRoot in Mathematica (Wolfram Research) computing environment (S1–S5 Appendix). Stability analysis was performed using eigenvalues computed on each steady state solution using the numerical continuation method matcont in Matlab (Mathworks, Inc.) (S1 Text). Eigenvalues of the associated ODE system are commonly used to assess the stability of each steady state and to identify critical points where the system enters a qualitative change. The sign of the real part of the eigenvalue indicates the stability of the steady state (i.e., negative eigen value implies stable steady state, whereas positive eigen value implies unstable steady state). A zero eigenvalue for the real part is associated to a critical point (indicated by LP in Fig 3) where the system changes from stable to unstable or from unstable to stable state.

## One-dimensional ODE representation of the reduced mFAO pathway

We considered a part of the mFAO pathway (Fig 3) involved in the conversion to and from C4-ketoacyl-CoA. In this pathway three CoA metabolites: C4-ketoacyl-CoA, C4-hydroxyacyl-CoA and acetyl-CoA and two enzymes: MSCHAD and MCKAT are involved. Moreover, competitive inhibiting substrates and products of various carbon chain length are involved as shown in Fig 3.

The ODE model associated with the pathway in Fig 3 for the rate of change in the concentration of C4-ketoacyl-CoA in the mitochondrion is given by

$$\frac{dM}{dt} = \frac{vMSCHADC4 - vMCKATC4}{VMAT},$$

where M denotes the concentration of C4-ketoacyl-CoA, VMAT is the volume of mitochondrion and the enzyme reaction rates vMSCHADC4 and vMCKATC4 are expressed as

$$vMSCHADC4 = \frac{vmschad \left( \frac{H4(N-N_D)}{K_{H4}K_N} - \frac{MN_D}{K_{H4}K_N K_{eqMSCHAD}} \right)}{\left( 1 + \frac{H4}{K_{H4}} + \frac{M}{K_{Msc4}} + \frac{Q_1}{K_{Q1}} \right)\left( 1 + \frac{N-N_D}{K_N} + \frac{N_D}{K_{N_D}} \right)} \tag{5}$$

$$vMCKATC4 = \frac{vmckat \left( \frac{M(T-S-M)}{K_{Mc4} K_{CoA}} - \frac{AcetylCoA\ AcetylCoA}{K_{Mc4} K_{CoA} K_{eqMCKAT}} \right)}{\left( 1 + \frac{M}{K_{Mc4}} + \frac{M}{K_{AcoA}} + \frac{Q_2}{K_{Q2}} \right)\left( 1 + \frac{T-S-M}{K_{CoA}} + \frac{AcetylCoA}{K_{ACoA}} \right)} \tag{6}$$

Details of notations and their descriptions are provided in the S1 Text.

At steady state

$$\frac{dM}{dt} = 0$$

implying

$$vMSCHADC4 - vMCKATC4 = 0. \tag{7}$$

Substituting (5) and (6) into (7), and with further algebraic simplification (S2 Text) provided a cubic polynomial of the form

$$aM^3 + bM^2 + cM + d = 0$$

where the coefficients a, b, c and d are functions of model parameters, moiety conserved quantities and concentrations of metabolites other than M in the model. Thus, in principle, solving the cubic polynomial (8) for the concentration M can provide one or three steady state solutions depending on the parameter values, moiety conserved quantities and concentrations of other metabolites in the mFAO model. The case of three real solutions, namely, three steady states that could lead to bistability, can occur only when $b^2 - 3ac > 0$, that is, if $c < 0$ or $c < b^2/3a$, since $a > 0$ as it is based on multiplications and additions of only positive constants (S2 Text).

## Supporting information

**S1 Fig. Indirect Substrate Inhibition of MCKAT.** (A) Artificially excluding substrate inhibition of C6-acylCoA from MCKAT resulting a minor shift in the bistability region to higher palmitoyl-CoA. (B) Artificially excluding substrate inhibition of C4-acylCoA from MCKAT resulting the model to exhibit a saturation kinetics and the bistability disappeared, i.e., the low-flux state disappeared at least up to 250 μM of palmitoyl-CoA.
(TIF)

**S2 Fig. Mitochondrial CoA concentrations in the modular analysis of mFAO model around C4-KetoacylCoA.** The plots show the depletion of free CoA and accumulation of C4 – C16 intermediate CoA esters with increasing C4-KetoAcylCoA.
(TIF)

**S3 Fig. NADH production and ACOT fluxes.** The left panel shows the NADH production with and without including ACOT in the model. The right panel shows carbon-chain specific activities of ACOT.
(TIF)

**S4 Fig. Ratio of NADH production flux to uptake flux at steady state.**
(TIF)

**S5 Fig. Mitochondrial CoA concentrations in the mFAO model with varying total CoA pool.** Total CoA pool (A) 2500 $\mu$M, (B) 3500 $\mu$M, (C) 5000 $\mu$M, (D) 7500 $\mu$M, (E) 10000 $\mu$M. Each plot shows as the concentration of palmitoyl-CoA increases, the depletion of free CoA and accumulation of intermediate CoA esters. The sharp decline in free CoA is shifted to higher palmitoyl-CoA concentration with increasing total CoA pool from 2500 to 10000 $\mu$M.
(TIF)

**S6 Fig. Magnified small bistability regions of varying NAD+:NADH and FAD:FADH$_2$ ratios (Fig 4 in the main text).**
(TIF)

**S7 Fig. Bistability in mitochondrial fatty acid oxidation with varying total pool of NAD$^+$ and NADH between 100 and 300 μM and fixed NAD+:NADH at 20.**
(TIF)

**S8 Fig. Bistability in mitochondrial fatty acid oxidation with varying Vmax of CPT1.** With decreasing Vmax of CPT1, bistability set in at a higher palmitoyl-CoA concentration.
(TIF)

**S9 Fig. Bistability in mitochondrial fatty acid oxidation with varying levels of short chain fatty acid (acetate).** With increasing levels of acetate from 0 to 5 mM, the bistability behavior and flux decline eventually vanished.
(TIF)

**S1 Text. Ordinary differential equations and rate equations in the extended mFAO models.**
(PDF)

**S2 Text. Analytical derivation of steady states in one dimensional ODE for a reduced mFAO model.**
(PDF)

**S1 Appendix. Analyzing origin and causes of bistability in mitochondrial fatty acid oxidation model.**
(ZIP)

**S2 Appendix. Bistability in mitochondrial fatty acid oxidation with varying some parameters.**
(ZIP)

**S3 Appendix. Bistability and mitochondrial fatty acid oxidation regulation by short chain fatty acids.**
(ZIP)

**S4 Appendix. Bistability and mitochondrial fatty acid oxidation regulation by p46Shc protein.**
(ZIP)

**S5 Appendix. Bistability and mitochondrial fatty acid oxidation regulation using ACOT as a safety valve.**
(ZIP)

## Author Contributions

**Conceptualization:** Ernst C. Wit, Barbara M. Bakker.

**Data curation:** Fentaw Abegaz.

**Formal analysis:** Fentaw Abegaz.

**Funding acquisition:** Ernst C. Wit, Barbara M. Bakker.

**Investigation:** Fentaw Abegaz, Barbara M. Bakker.

**Methodology:** Fentaw Abegaz, Ernst C. Wit, Barbara M. Bakker.

**Project administration:** Ernst C. Wit, Barbara M. Bakker.

**Resources:** Ernst C. Wit, Barbara M. Bakker.

**Software:** Fentaw Abegaz, Anne-Claire M. F. Martines.

**Supervision:** Ernst C. Wit, Barbara M. Bakker.

**Visualization:** Fentaw Abegaz, Marcel A. Vieira-Lara.

**Writing – original draft:** Fentaw Abegaz, Anne-Claire M. F. Martines, Marcel A. Vieira-Lara, Melany Rios-Morales, Dirk-Jan Reijngoud, Ernst C. Wit, Barbara M. Bakker.

**Writing – review & editing:** Fentaw Abegaz, Anne-Claire M. F. Martines, Marcel A. Vieira-Lara, Melany Rios-Morales, Dirk-Jan Reijngoud, Ernst C. Wit, Barbara M. Bakker.

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
