## [Decision Letter · Decision Letter 0]

19 Feb 2021

Dear Dr. Bakker,

Thank you very much for submitting your manuscript "Bistability in Fatty-Acid Oxidation resulting from Substrate Inhibition" for consideration at PLOS Computational Biology.

As with all papers reviewed by the journal, your manuscript was reviewed by members of the editorial board and by several independent reviewers. In light of the reviews (below this email), we would like to invite the resubmission of a significantly-revised version that takes into account the reviewers' comments.

We cannot make any decision about publication until we have seen the revised manuscript and your response to the reviewers' comments. Your revised manuscript is also likely to be sent to reviewers for further evaluation.

Sincerely,

Anders Wallqvist

Associate Editor

PLOS Computational Biology

Jason Papin

Editor-in-Chief

PLOS Computational Biology

Reviewer's Responses to Questions

**Comments to the Authors:**

Reviewer #1: In this article, Abegaz et al. perform bi-stability analysis of a mathematical model describing beta-oxidation of fatty acids in rat-liver mitochondria. The article lays out the rationale for performing the analysis and then investigates the effect of specific parameters, such NAD/NADH ratio, on the observed bi-stable region of the model. The authors have assumed a fixed NAD/NADH ratio to avoid its effects on their model calculations. Specifically, the authors have chosen the NAD/NADH ratio of 20, which is twice the ratio reported by Williamson et al. (PMID: 4291787) and Henley et al. (PMID: 4312780) in rat-liver mitochondria. This is a major drawback of their model observations because the bi-stability phenomenon would not exist if they have chosen the ratio reported in Williamson et al. or Henley et al. Moreover, the authors found that hysteresis was observed with increasing concentrations of Palmitoyl-CoA; full oxidation of Palmitoyl-CoA produces 7 molecules of NADH and acetyl-CoA entering the TCA cycle generates 3 NADH molecules. Therefore, I suggest that the authors discuss the consequences of having a fixed NAD/NADH ratio on their modeling predictions and how their model results may be affected with a dynamic NAD/NADH ratio. In addition, I have a few minor comments:

1. Line 224-226: “This can be seen easily from Figure 4B: outside the stable points the system will move towards a higher C4-ketoacyl-CoA if the production by C4-MSCHAD is faster than its consumption by C4-MCKAT, and vice versa.” Outside of the stable point ‘S3’ C4-ketoacyl-CoA seems to be decreasing in Figure 4B -- it’s not clear what the authors are saying.

2. Line 231, “S0 in Figure 3C.” I think, the authors mean “Figure 4C.”

3. Line 325, “the rate v of enzyme i.” I think, the authors mean “the rate vi of enzyme i.”

4. Line 344, “[NAD+]:[NAD]” should be replaced by “[NAD+]:[NADH].”

Reviewer #2: This is a well-written paper about a computational model for the mitochondrial fatty acid oxidation (FAO) pathway. The work extends upon a previously published model (Ref 40) that identified medium chain 3-keto-acyl-CoA thiolase as a key enzyme in the pathway with regards to limiting flux. The current paper examines that effect in further detail and explores some interesting ways to increase flux through the pathway, which would be physiologically beneficial and clinically relevant if possible.

This reviewer is an expert in FAO physiology, and not computational biology. The following criticisms are raised from the perspective of relating the model back to physiology.

Major Criticisms

1. The model does not include a component related to carnitine availability or the role of the reverse carnitine cycle in maintaining intra-mitochondrial levels of CoA. The enzyme CrAT can convert short to medium-length acyl-CoAs (up to 8 carbons) into acylcarnitines and free CoA. The acylcarnitines are released. Similarly, CPT2 can operate in reverse if longer-chain acyl-CoAs accumulate. This should be discussed and incorporated into the model if possible.

2. The section on modeling of malonyl-CoA effects (beginning line 295) ends with the statement "Due to the stretching of the high-flux branch, malonyl-CoA can therefore activate the mFAO flux even though it acts through inhibition of one of its enzymes." The physiological relevance of the malonyl-CoA modeling is unclear. Under what circumstance would you have both high malonyl-coa and high levels of long-chain fatty acids? It is doubtful that malonyl-CoA would ever activate mFAO flux in vivo.

3. In the section on acyl-CoA thioesterases as a safety valve (beginning line 378), is the presence of ACOT2 already accounted for in the computational model? The model is based on liver mitocondria, which possess the thiolase ACOT2 as described in Ref 64. Is the model adding addtional thiolase activity? This was unclear to me. Further, it is stated "However, excessive accumulation of intracellular acyl-CoAs can be toxic in itself and also limit the mFAO flux due to CoA depletion." But free fatty acids are also toxic, which should be discussed. Further, liver possesses abundant medium-chain acyl-CoA synthases inside the matrix. In the model, would adding a medium-chain thiolase merely a create a vicious cycle of cleaving medium-chain acyl-CoAs and re-esterifying them to CoA at the expense of matrix ATP?

Reviewer #3: General

The presented manuscript “Bistability in Fatty-Acid Oxidation resulting from Substrate Inhibition” provides by kinetic modeling and mathematical analysis the prediction of bistability in fatty-acid oxidation, which is characterized by the coexistence of a stable low and high-flux state of fatty-acid oxidation. The mathematical analysis is original, the manuscript well written and is of substantial interest for theoretical as well as experimental research devoted to fatty-acid metabolism. Despite that, I advise to strengthen the prediction of bistability by a comparison to experimental data and observations as well as suggestions how to perform an experimental validation. In addition, I suggest below some major and minor remarks and suggestions to improve this manuscript.

Major

• Link or releavance of bistability under physiological conditions. I am missing throughout the text the effort to convince the reader that the bistability is present under physiological conditions. Perhaps, gene expression data or any other experimental observations are only explainable by the coexistence of a high and low flux state? Further, how do you suggest to experimentally verify your predictions and which systems could be used (human studies, animal models, tissue cultures, cell cultures)?

• Enzyme concentration is fixed during simulation (Vmax = kcat*E). Cell could alter “E” by transcription etc., how would this change bistability? Is it possible to alter enzyme abundance to prevent the low flux state or make a transition from low to high flux state?

• Analytic analysis

o Not ideal to start with a simplification before any result of the full model is presented

o I respect the effort to provide an analytic solution. However, the result “can in principle exhibit bistability” is rather weak in comparison to the extensive derivation shown in the supplement

o Suggestion: start with a general results and features of the model (see below) and provide then the analytic solution of the simplification

• Possible additional analyses:

o dynamic of system at non steady state (oscillation? Time until steady state is reached? …)

o Overall reaction, elementary mode, energy flux/yield of metabolic system (with and without changes in this manuscript)

Minor

• Provide Mathematica Notebook files (.nb) also as PDF to make it easily accessible for readers with no Mathematica license

• Line 60-61: Could you elaborate further, how FAS prevents substrate inhibition? Isn`t the retainment of intermediates a form or sign of substrate inhibition?

• Line 72: Is the strategy to artificially increase mFAO and energy turn over in general a therapeutic option without draw back? I could imagine the risk of cell damage and DNA mutagenesis by increased ROS production etc., please consider and discuss possible side effects.

• Line 74-81: I think some references could be added here to back the statements.

• Line 145-146: the main derivation is found in the supplement; please add this as additional reference.

• Line 159-160: It is not clear how the and when the palmitoyl-CoA concentration is changed. Is the concentration changed over time, simulation started with different initial conditions, …?

• Line 234: “bistability originates from …” I think the statement that one could define the origin of bistability to one feature is to strong. The analysis shows that is a necessary condition as well as other parameters or parameter combination. I would advise to rephrase this and be aware in the discussion of results that there is no single origin of bistability in such models, but rather a combination of topology and kinetics.

• Line 259-275: I understand that the NAD/NADH ratio is important. But, is the overall concentration of NAD+NADH as well an important factor for bistability? And while varying NAD/NADH ratio, was the sum NAD+NADH kept constant? Please elaborate.

• Line 300-301: “The maximum flux was the same” So, it is a competitive inhibitor?

• Line 295-307: What are physiological expected concentrations of malonyl-CoA? The higher stretch of low and high-flux state would be physiologically unfavorable, or not? Could you elaborate on this?

• Line 325: Which parameter p have been used to calculate the control coefficients shown in Figure 6? If all, was an average calculated?

• P46Shc and Shc: Could you elaborate further on other functions of p46Shc and Shc in general (human, mice, mammals)? Are there any harmful side effects if this is depleted? It sounds too good to be true if a knock out of Shc solves the problem.

• Influence of Acetate. In Figure 7 and in the text you show that there is no bistability with acetate = 3mM. Is there any concentration of acetate >0 where there is still bistability?

• Relation of results to diets and eating regimes: If I understand the results correctly, your results confirm the observation that fasting periods and less frequent meals per day is better. By big variations of fatty acid supply and oxidation supply, the risk of being stuck in the low flux state would be minimized, isn’t it?

• Line 544: the letter N is here used for the number of metabolites. Later in equation (5) it is used again, but I think here it represents the concentration of NADH/NAD.

• Equation (4): There is no explanation of “sf_vmckat” and its meaning.

• Line 568: “a few metabolic intermediates were set to be constant” Again, I think it is crucial to explicitly state which metabolites are external and constant to understand the steady state flux properly.

• Line 617: a>0 is crucial and really trivial to see. E.g. if a=0, it is a quadratic function with maximal two steady state and hence no bistability. Please elaborate shortly, why a>0 is justified.

• Figures:

• Figure 1, please add some indication of external (constant or conserved moieties) metabolites. Also elaborate this in the text. Otherwise, the steady state of the metabolic system is not understandable.

• Figure 2: there is no Figure 2?

• Figure 4: Panels B and C are a bit confusing. It is not really clear what is shown by the lines (mathematically). Please indicate this in the caption or better find overall a better visualization. Are the plots on the right hand side of panel B and C for a fixed palmitoyl-CoA concentration?

• Figure 5: For all panels I suggest to use a color gradient for the lines indicating the varied parameter value instead of the high number of colors. In the current version one cannot see intuitively the relation of increasing the parameter value and the corresponding lines. Further, in 5B consider to flip the ratio FAD/FADH2 to FADH2/FAD and use the same ratios as in 5A to have direct comparison of the impact on bistability.

• Figure 6: I would suggest to use letters (A,B) again for the subgraphs instead of left/right.

• Figure 7: I suggest to rearrange the subpanels to A, B (first row) and C, D second row. I think this is the typical reading pattern across journals. Also check that font sizes of axis titles etc. are uniform, it looks a bit messy at the moment.

•

• References:

• Check overall citation formatting (species name italics, capitalization)

• [20] and [37] are duplicates

• Wording:

• Line 45, “network in -> of? Escherichia coli”

• Line 54, “loop is born” sounds not very scientific

• Line 87, “parameters for -> of? rat-liver”

• Line 159 “parameters [were] unchanged”

• Line 210: “was fixed and varied” sounds like a contradiction (constant over time, but varied initial concentration)

• Line 264: “[link to] control [of] cell metabolism”

• Line 268: “For [a] small value”

• Line 285: “Bistability existed” -> “exists”?

• Line 364-365: “activate the mFAO flux by […?] and”

• Line 431: rephrase “a stable state flux was found either at high or at low flux state” -> e.g. “a stable state flux with either low or high flux”

• Line 434: “de novo” italic?

• Line 442-444: Somehow this sentence is broken at “… coexist which state …”

• Line 498: “… higher FADH2 [concentration] …”

• Line 554: avoid “ … and … and … “

**Have all data underlying the figures and results presented in the manuscript been provided?**

Reviewer #1: Yes

Reviewer #2: Yes

Reviewer #3: Yes

PLOS authors have the option to publish the peer review history of their article (what does this mean?). If published, this will include your full peer review and any attached files.

Reviewer #1: No

Reviewer #2: No

Reviewer #3: No
---

## [Decision Letter · Decision Letter 1]

7 Jul 2021

Dear Dr. Bakker,

We are pleased to inform you that your manuscript 'Bistability in Fatty-Acid Oxidation resulting from Substrate Inhibition' has been provisionally accepted for publication in PLOS Computational Biology.

Best regards,

Anders Wallqvist

Associate Editor

PLOS Computational Biology

Jason Papin

Editor-in-Chief

PLOS Computational Biology

Reviewer's Responses to Questions

**Comments to the Authors:**

Reviewer #1: My comments have been addressed.

Reviewer #2: The revisions made to the manuscript have satisfied my concerns from the original review.

Reviewer #3: I appreciate very much the authors efforts to improve the manuscript. In my opinion they nicely incorporated my suggestion and comments of the other reviewers. To this end, I have no further objections before publication.

Thank to the authors for the detailed response and I congratulate to the substantially improved manuscript.

**Have the authors made all data and (if applicable) computational code underlying the findings in their manuscript fully available?**

Reviewer #1: None

Reviewer #2: Yes

Reviewer #3: Yes

PLOS authors have the option to publish the peer review history of their article (what does this mean?). If published, this will include your full peer review and any attached files.

Reviewer #1: No

Reviewer #2: No

Reviewer #3: No

---

## [Editor Report · Acceptance letter]

4 Aug 2021

PCOMPBIOL-D-21-00137R1 

Bistability in Fatty-Acid Oxidation resulting from Substrate Inhibition

Dear Dr Bakker,

I am pleased to inform you that your manuscript has been formally accepted for publication in PLOS Computational Biology. Your manuscript is now with our production department and you will be notified of the publication date in due course.

With kind regards,

Melanie Wincott
